# Next-generation microbiology: from comparative genomics to gene function

Carolin M. Kobras[1]  , Andrew K. Fenton[1*]   and Samuel K. Sheppard[2*]

* Correspondence: A.K.Fenton@
sheffield.ac.uk; S.K.Sheppard@bath.
ac.uk
[1]Department of Molecular Biology &
Biotechnology, University of
Sheffield, The Florey Institute for
Host-Pathogen Interactions,
Sheffield, UK
[2]Department of Biology &
Biochemistry, University of Bath,
Milner Centre for Evolution, Bath,
UK

## Abstract

Microbiology is at a turning point in its 120-year history. Widespread next-generation sequencing has revealed genetic complexity among bacteria that could hardly have been imagined by pioneers such as Pasteur, Escherich and Koch. This data cascade brings enormous potential to improve our understanding of individual bacterial cells and the genetic basis of phenotype variation. However, this revolution in data science cannot replace established microbiology practices, presenting the challenge of how to integrate these new techniques. Contrasting comparative and functional genomic approaches, we evoke molecular microbiology theory and established practice to present a conceptual framework and practical roadmap for next-generation microbiology.

## Introduction

Experimental approaches for studying bacteria have changed dramatically over the last 20 years [1]. Shifting in response to public interest and fuelled by technological advances, understanding of these remarkable organisms continues to rapidly advance. We now know more than ever before about the metabolism, environmental context and host interactions of microbes, and the rate of discovery shows little sign of slowing. Among the most influential shifts in technology has been the increasing use of large sequencing datasets in research practice. These contemporary research approaches continue to gain momentum, expanding into ever more ingenious ways of using sequencing data to discover complex patterns of behaviour and reach a deeper understanding of the bacterial cell. This rapid advancement has many conceptual benefits but has also come at a significant cost, as laboratories struggle to integrate these techniques and apply best research practices to new types of data.

The magnitude and complexity of large sequencing datasets can make them appear abstract to the non-specialist, potentially leading to subjective judgements about whether to believe the analyses or not. This can, in turn, risk general disenfranchisement of microbiology researchers away from genomic data, promoting an over-reliance on outside proofs or validations to give meaning to sequencing-based datasets. Here, we argue for an integrated future for microbiology that combines the strengths

of traditional microbiology with the promise of emergent sequencing technologies. Addressing the widening gap in research practice, we discuss some of the most influential methodologies, the validation of findings from large sequencing datasets, and how comparative and functional genomics can be integrated to advance microbiology from fundamental discovery to contemporary microbiology research practice.

## Using a data deluge for qualitative and quantitative microbial genomics

It has been well over a decade since next-generation sequencing (NGS) platforms became widely available for microbial genomics. The cost of sequencing has continued to fall to a point where large sequence datasets are within the budget of most research groups. This democratisation of technology was not driven by a fundamental change in how DNA is sequenced. In fact, the major shift came through the upscaling of bridge amplification in the Illumina sequencing-by-synthesis process [2, 3]. This allowed the simultaneous sequencing of millions of individual DNA molecules in parallel by NGS machines generating huge amounts of data. While new single-molecule sequencing technologies developed by Oxford Nanopore and Pacific Biosystems gather momentum [4], the massively parallel Illumina NGS approach remains a major driver in the generation of large-scale DNA sequencing datasets. Key to the widespread use of NGS methodology are the diverse applications. Broadly, the functionality can be described under two contrasting modes. The first is a high-accuracy DNA sequencing function best applied on either de novo genome sequencing or making detailed comparisons between genomes. Here, the huge numbers of individual sequencing reads are combined to remove errors in base calling and generate high-confidence ensemble averages. The second mode is a counting function used to survey mixed populations of DNA or RNA molecules. Here, each sequencing read is examined individually, separated into groups and scored. This approach can, for example, be applied to measure the relative frequencies of mRNA levels in a cell or to capture the composition of a bacterial population from an environmental sample.

## Transformative sequencing technologies and the genetics of phenotype variation

Determining the genetic basis of phenotype variation is among the most pervasive aims in microbiology. This is a major challenge and requires understanding of how changes to genes, and their constituent DNA sequences, can alter gene function and affect a phenotypic change over time. Two of the most transformative techniques that address this in bacteria are genome-wide association studies (GWAS) [5–7] and transposon insertion sequencing methods (here referred to as Tn-Seq, but also known as HITS, InSeq or TraDIS) [8–11]. Both techniques are powered by NGS, but each uses different functions of DNA sequencing technologies. GWAS requires genomes from multiple strains within a population to identify genomic elements that are statistically associated with a given phenotype or environmental condition [12, 13] and therefore uses the high-accuracy function of NGS. In contrast, Tn-seq profiles fewer strains and uses the DNA counting function to identify transposon insertions in populations of mutants to identify the contribution each gene makes to bacterial survival within the specific experimental context [14, 15].

### Describing population-wide genomic variation

The availability of numerous high-quality bacterial genomes representing the extraordinary complexity of phenotypic and genotypic variation in natural populations has inevitably led microbiologists to new analytical techniques. Drawing on methods that were pioneered in human genetics, early bacterial GWAS approaches [5] have been adapted to become an important in silico tool for population-wide genomic screening [16]. Studies typically involve sampling and genome sequencing of hundreds of isolates from different environments or conditions and identifying genetic elements (e.g. single nucleotide polymorphisms (SNPs), k-mers or accessory genetic elements) that are significantly associated with a phenotype in question (Fig. 1). Now widely used, bacterial GWAS have successfully identified candidate genes involved in host specificity [5, 17], virulence [6, 18–24], the duration of pathogen carriage [25], and antibiotic resistance [7, 26–29].

The widespread application of bacterial GWAS has been made possible by adapting the methodological and analytical assumptions of human GWAS in two important ways. First, bacterial GWAS not only targets homologous sequence variation but also aims to identify the numerous accessory genetic elements and genes that may be found in some, but not all, isolate genomes [5, 30]. Second, and most importantly, it accounts for the strong linkage disequilibrium resulting from the clonal mode of bacterial reproduction. Accounting for this population structure is particularly important when considering the genetics underlying phenotype variation as causal variants will be co-inherited with linked loci that may have no adaptive function [12, 13]. In highly structured bacterial populations entire clusters of strains may share elements that have facilitated their expansion as well as those that simply reflect common ancestry. To address this, population subsampling [20, 31], linear mixed models [27, 32] and phylogenetic trees [21] can be incorporated into analyses to account for the clonal frame of the population. Resultant associations that cannot be explained by the effect of shared ancestry can represent convergent genomic signatures in groups of divergent strains. This provides clues to the evolutionary forces acting on the bacterial genome.

Sophisticated bioinformatics analyses of ever larger genome collections are: (i) incorporating quantitative trait variation [28]; (ii) conditioning on multiple genomic or phenotypic determinants [20, 29]; (iii) using machine learning to quantify the relative importance of associated elements in explaining the observed phenotype [20, 31, 33]. However, while bacterial GWAS approaches benefit from retaining the natural population setting of a given phenotype, they often return many thousands of genetic elements associated with complex traits such as host association or virulence [34]. In such cases, it can be extremely difficult to identify the role of individual genes and unravel the myriad interacting selective effects that shape the observed genomic variation. For this it may be necessary to move beyond in silico statistical associations and understand the function and importance of specific genes under more carefully controlled conditions.

### Studying gene function through modification and inactivation

Observing how small genomic differences, in otherwise isogenic strains, influence the phenotype provides evidence about the functional consequence of sequence variation.

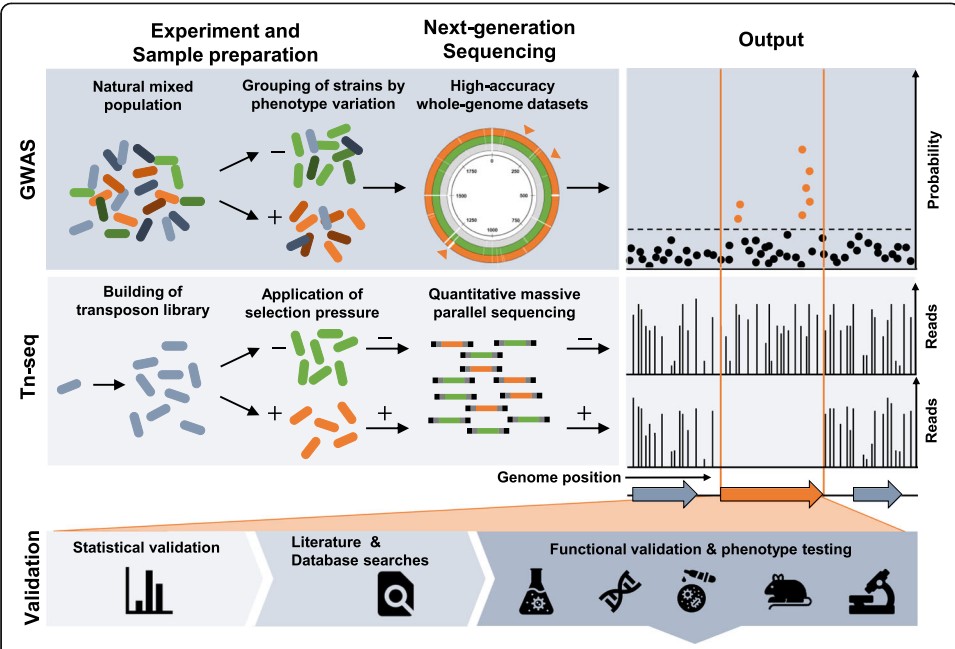

**Fig. 1** | Schematic overview of the GWAS and Tn-seq methods and a generalised validation pipeline. The gene highlighted in orange represents an idealised output for each approach. GWAS panels: In general, samples used for GWAS studies are directly isolated from the environment of interest. The phenotype of each isolate is tested and/or recorded, before whole-genome sequencing. Correlations between changes in observed genotypes and phenotype variations are determined. The output of GWAS can be displayed as a Manhattan plot, with the probability that each genetic variant detected in a population is associated with the phenotype of interest plotted against the genome positions. If variants fall above a certain probability threshold (dotted line), they are considered associated with the phenotype of interest (points highlighted in orange). Tn-seq panels: Saturated transposon libraries are grown in the presence and absence of the selection pressure of interest. Transposon-genome junctions from each member of the library are amplified and sequenced. Exploiting the quantitative function of massive parallel sequencing, the number of reads found for each transposon insertion junction are plotted against the genome position. The datasets obtained from libraries with and without the selection pressure are then compared to identify the contribution of each gene to the fitness. Areas of the genome with a different pattern of transposon insertions are deemed to be associated with the selection conditions (see region within the orange box). Validation panels: Initially the results of both methods are validated statistically and first insights into gene function are gained through literature and database searches. Deeper studies confirm the genotype-phenotype relationship of the results with a functional validation in the laboratory using a variety of experimental approaches

Over decades, microbiologists have identified the function of numerous genes across multiple species mainly through investigating the effect of gene loss. It is possible to infer gene function by inactivating specific genes, usually through introduction of a specific mutation into the genome of an organism and comparing the resultant phenotype to that of a 'wild-type' strain. While this is relatively laborious compared to observing genomic variation in natural populations *in silico*, it provides much greater control of the genomic variation and the conditions in which the gene function is being tested. Extending the principle of gene inactivation for genome-wide functional studies, ordered gene deletion libraries have been generated for several model laboratory strains [35–38]. In these libraries, all non-essential genes have been disrupted by the insertion of antibiotic markers, allowing the rapid screening of phenotypes under different selective conditions. Furthermore, random chemical or UV mutagenesis have been used to generate ordered mutant libraries, without requiring any a priori genetic manipulation

of the bacterial strain [39]. While this allows investigations into bacterial species that are hard to manipulate genetically, it may be difficult to generate sufficient mutations for complete gene coverage in the screen, particularly as the whole genome of each strain needs to be sequenced to locate a mutation.

### Whole-genome fitness profiling using transposon insertion mutagenesis

Building on the concept of using large-scale gene deletion libraries that cover the entire genome of the bacterium, a revolution in these methodologies began just over a decade ago with the integration of quantitative high-throughput NGS technologies that capture the complexity of a large transposon-insertion library in one sequencing step [14, 15]. Developed around the same time, conceptually similar techniques including Tn-seq [8], TraDIS [9], HITS [10], and INSeq [11], all use large transposon insertion libraries, across which all or most non-essential genes contain transposon insertions. Selection pressure is applied to these library strains by growing them in defined in vitro or in vivo conditions (Fig. 1). Subsequent amplification and sequencing of the transposon-genome junctions in the libraries allows the insertion location of each transposon to be determined for each condition. The key feature of these approaches are the resulting 'profiles' of transposon insertions that reflect the fitness contribution each gene had under the selective conditions of the experiment. Specifically, regions of the genome where transposon insertions are statistically underrepresented likely contain genes that are essential for the bacteria to survive in the experimental conditions [8, 40–42].

This whole-genome fitness profiling method has linked many genes with metabolic pathways [43] and important phenotypes including stress response and antibiotic resistance [44–46], virulence and survival in the host environment [9–11, 47–51]. Furthermore, by deleting specific query genes it may be possible to identify gene interactions [8, 52, 53] and to examine the role of non-coding and regulatory DNA [54]. Most Tn-seq approaches rely on negative selection via gene inactivation. However, transposons carrying outward facing promoters can result in the upregulation of neighbouring genes. This allows controlled analysis of functional gene upregulation, an approach applied to the study of antibiotic resistance [55–57].

Recently, CRISPR interference (CRISPRi)-based methods have been added to the assortment of functional genomic tools [58]. Here, a small guide RNA forms a complex with the inactivated DNA-binding protein Cas9 and together they bind a specific region of the genome. The complex blocks RNA polymerase at the targeted site through steric hindrance, and represses transcription of the targeted gene [59, 60]. In a genome-wide screen, large libraries of mutants, each containing a different CRISPRi construct, can be captured by high-throughput NGS [61–66]. In contrast to Tn-seq, CRISPRi libraries have the potential of covering all genes in a genome, including essential genes. However, secondary and off-target effects still have to be carefully considered.

### Understanding bacteria in the wild

When trying to understand the genomics underlying trait variation in bacteria microbiologists must make compromises. The major challenge is to balance the experimental control needed to understand the function of specific genes with the requirement for data that is relevant in natural populations. This is illustrated by contrasting the

selective conditions in Tn-Seq and GWAS approaches (Fig. 2). By deliberately limiting the selection pressures, Tn-seq studies provide a clear path to the functional validation of genes, often involving recreation of the initial experimental conditions used for the Tn-seq screen and measuring the fitness of genetically modified bacteria in competition-based assays [8, 44, 67]. However, in some cases there is surprisingly little overlap among the genes required for growth in particular conditions when comparing datasets between different laboratories [68]. This may be because of differences in: the precise experimental conditions; the transposons used; false-positives resulting from polar effects of transposon insertion; library selection and handling methods [69]. While the high-throughput nature of these methods, and appropriate validation, can largely overcome the challenge of reproducibility, the major strength of Tn-seq can also be considered its limitation. Specifically, while the deliberate constraint of the selection conditions acting on the bacteria facilitates functional genomics, these studies can also be criticised for lacking 'real-world' insight into genotype-phenotype relationships.

To enhance the relevance of laboratory findings for natural bacterial populations, recent multi-strain Tn-seq studies have included clinical or environmental isolates [42, 44, 46, 57]. However, there is a clear benefit to inference from bacteria in the wild. In this respect, GWAS is a powerful approach, as it directly surveys natural genotype-phenotype associations. This inevitably means that bacteria are sampled from dynamic systems and will have been exposed to a complex set of selection pressures, not all of which are directly related to the primary condition of interest. Structured sampling, replication and statistical tests, and in silico validations can strengthen assumptions about causal genetic variations [20], but recreating the experimental conditions in a laboratory setting may be extremely difficult, leading to difficulty when trying to understand the value of GWAS for lab-based microbiologists.

The complementary strengths and limitations of population-wide screens and laboratory fitness profiling methods provide a means to identify the genetic basis of complex bacterial traits (Fig. 2). Therefore, in combination, techniques such as GWAS and Tn-seq could provide insights into the behaviours of bacteria in natural environments in

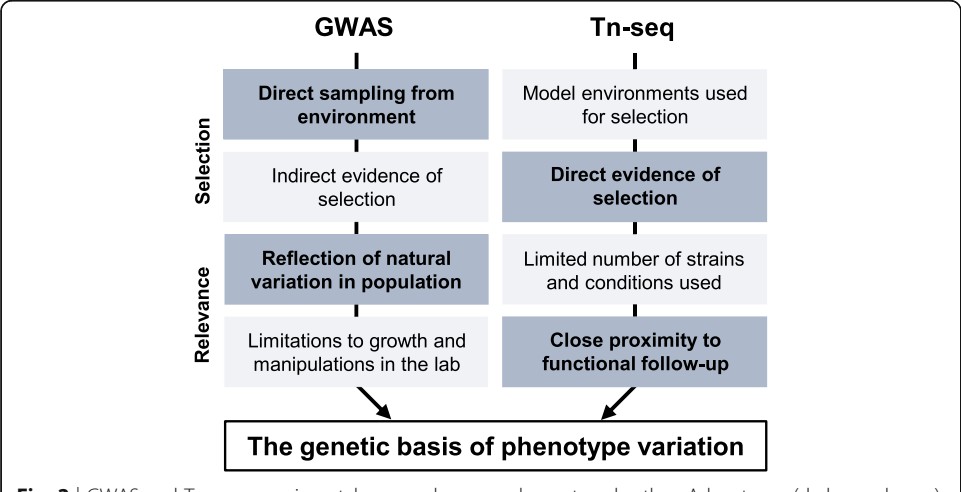

**Fig. 2** | GWAS and Tn-seq experimental approaches complement each other. Advantages (dark grey boxes) and limitations (light grey boxes) of both experimental approaches, focusing on differences in the application of selection pressure and relevance of the information gained from both methods

such a way as to be experimentally tractable in a laboratory setting. The initial steps taken to ensure data quality for both methods are similar, involving technical repeats and statistical validation. Yet, it is the experimental proof that specific genetic variants cause observable phenotypes that makes these studies so impactful. The challenges are how to achieve these proofs, what experimental methods should be used, and against what guidelines might we measure the evidence.

### Functional validation in the post-genomics era

To the data analyst, functional genomic inference may be considered 'validated' if associations are proven robust against a series of statistical challenges. However, for laboratory-based researchers, in silico findings are typically considered 'validated' only when their effects can be reproduced using a complementary experimental approach. This requirement for experimental reproducibility has been a central tenet in microbiology since the publication of Koch's postulates [70]. Adapting this conceptual framework in 1988, Stanley Falkow established a set of rules to prove causality of molecular genetic changes to disease phenotypes (Fig. 3) [70–72]. Subsequently adjusted to fit different research areas [73–76], these Molecular Koch's postulates remain engrained in molecular microbiology best practice because of the scientific rigour they promote.

A major limitation of population-scale and genome-wide genetic screens is that a relatively small fraction of candidate genes are functionally validated. In some cases, in silico and laboratory-based genomic screens have employed follow-up gene inactivation and phenotype investigations to link the function of specific genes to pathogenicity [6, 18, 20, 31, 48, 77], survival and transmission [17, 19, 51] and antimicrobial resistance [29, 44, 57]. This experimental confirmation can be a challenging task given the large numbers of genes involved in complex phenotypes but it remains important for robust genotype-phenotype association. To address this we propose revised Molecular Koch's postulates for functional genomic validation of NGS analyses (Fig. 3).

| Molecular Koch's postulates | Next-generation Koch's postulates |
|---|---|
| 1. The phenotype or property under investigation should be associated with pathogenic members of a genus or pathogenic strains of a species. | 1. **The genetic variant under investigation should be significantly associated with the phenotype.** Preferably a strong association, which is true across several bacterial strains or species. |
| 2. Specific inactivation of the gene(s) associated with the suspected virulence trait should lead to a measurable loss in pathogenicity or virulence, or the gene(s) associated with the supposed virulence trait should be isolated by molecular methods. Specific inactivation or deletion of the gene(s) should lead to loss of function in the clone. | 2. **Specific recreation of the genetic variant associated with the suspected phenotype should lead to a measurable change thereof.** In most cases this is best met through gene inactivation. In the case of an essential gene, a method of depletion is often used to investigate changes in phenotype. |
| 3. Reversion or allelic replacement of the mutated gene should lead to restoration of pathogenicity, or the replacement of the modified gene(s) for its allelic counterpart in the strain of origin should lead to loss of function and loss of pathogenicity or virulence. Restoration of pathogenicity should accompany the reintroduction of the wild-type gene(s). | 3. **Reversion or allelic replacement of the genetic variant should lead to restoration of the phenotype.** |

**Fig. 3** | Molecular Koch's postulates, including a generalised revision for the purposes of this review. Stanley Falkow's adaptation of Koch's postulates (left) have been the gold standard to support causal links between genotypes and bacterial phenotypes for decades [71, 72]. To provide wider accessibility and application to microbiology research more generally, we have revised these postulates (right). Importantly, these postulates are adapted to guide functional validation and the authors acknowledge the full set might not be fulfilled in all cases

In most cases, genomic screening approaches will automatically fulfil the first postulate - that bacterial strains with an identified genetic variant should display the phenotype of interest (Fig. 3). Ideally, the relationship between genetic change and phenotype of interest would be as direct as possible, shifting only one experimental variable at a time and showing a large effect size. In this way, focusing on strong correlations can be useful as it provides the best experimental proofs.

According to the second postulate, specific changes made to the gene of interest should result in a change to the phenotype in question (Fig. 3). While not all genetic variants will result in the loss of the gene function, the generation of targeted gene deletions and reproduction of the experimental conditions leading to the expected phenotype is an approach often used to meet this postulate. To achieve this, tools for marked and markerless gene deletions have been developed [78–80], with the availability of ordered transposon or single-gene deletion libraries expediting this process for some bacterial species [35, 38, 81]. In cases where a gene is essential for the survival of a bacterium, depletion systems can be used but often require an established set of genetic tools to function. A popular example for sequence-specific repression of gene expression is CRISPRi [59, 60].

Finally, the third postulate focusses on restoring the observed phenotype through genetic complementation (Fig. 3). This is an essential step to close the loop and prove causation but may require a more sophisticated set of genetic tools to achieve. The most direct example is complementation via a conditional expression system, usually on a plasmid or at an ectopic locus in the genome. Alternatively, more subtle genetic manipulates can be used to meet this postulate, for example the introduction of a base pair change into the genome that complements the phenotype, an approach often achieved by site-directed mutagenesis [82].

The functional validation of genomic screens, potentially based on these revised Molecular Koch's postulates, provides an ambitious target. In fact, it may be extremely difficult in practice to identify, remove and reinstate the genetics underlying trait variations. For example, where multiple independent variations cause subtle phenotypic changes or where genes are part of interactive networks and co-vary because of epistasis [83, 84]. As such, Molecular Koch's postulates exist principally as a 'gold standard' rather than a definitive list of experimental criteria. In practice, integrated microbiology should layer multiple experimental approaches, using the minimum number of methods required to meet the burden of proof and focus validation effort for functional follow-up studies.

### A roadmap for next-generation functional microbiology

Identifying and then proving that a genetic variant causes a phenotypic change is a considerable step towards understanding bacterial genomics. However, determining the precise role of the gene and how the encoded protein may function requires further study. This can be challenging, particularly for researchers with no background in molecular microbiology, even when the genetic methods or biochemical assays require fairly basic laboratory equipment. Therefore, just as laboratory microbiologists are encouraged to embrace contemporary genomic approaches, so bioinformaticians might consider the central role of functional microbiology in various ways (Fig. 4).

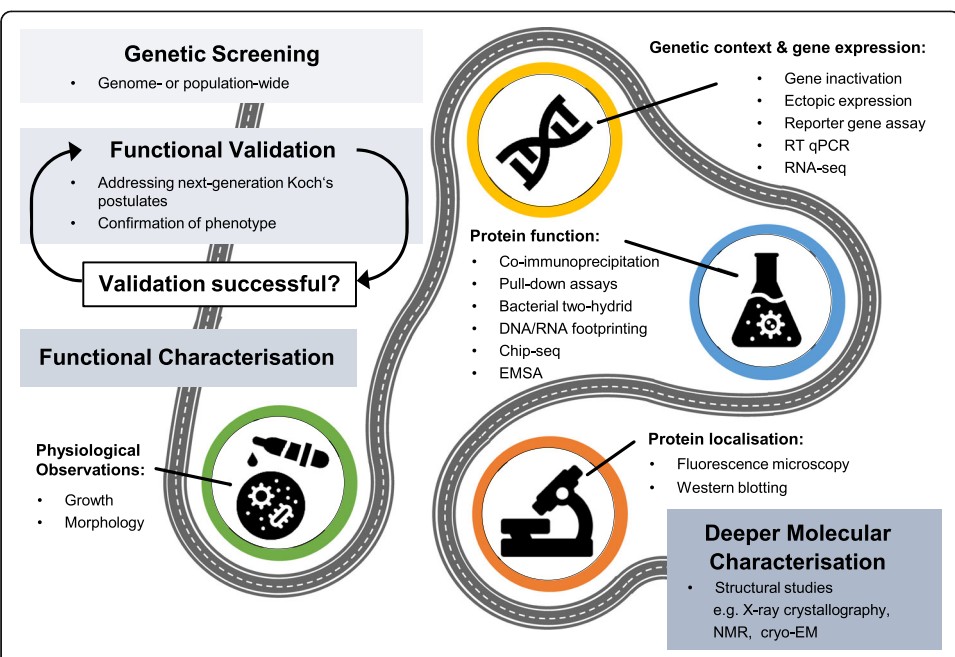

**Fig. 4** | A roadmap to understanding gene function. This figure splits the pathway of identifying, validating and investigating gene function into three different parts, with increasing depth of understanding. For the kinds of screening approaches discussed in this review, functional validation (top panel) is a crucial step in confirming the link between candidate genes and phenotype, which we argue should be carried out against criteria set out in next-generation Koch's postulates. If successful, genes should be further functionally characterised towards deeper understanding (middle panel). This can be achieved using some of the methods laid out in the central panel and are shown here to guide researchers who are less familiar with these approaches. Deeper characterisations often require more specialist equipment and may be beyond the scope of non-specialist labs, we highlight a few examples here to place these types of methods in context (bottom panel)

### Physiological observations

Although considered rather simplistic, basic physiological observations are an important starting point for studies of gene function in microbiology. Differences in growth rate, signs of growth arrest, and early lysis are often tell-tale signs and should not be overlooked. Live-cell microscopy can also provide insights into whether a gene of interest affects cell morphology.

### Genetic context and gene expression

In bacteria, genes with related functions often cluster together in operons. Therefore, deciphering the genetic context of candidate genes, as well as understanding where and when a gene is expressed in a bacterial cell, can provide insights into how they function. Determining the pattern and timing of gene expression can be accomplished by replacing the coding sequence of candidate genes with a reporter construct, allowing expression to be monitored through fluorescence, luminescence or enzymatic activity [85]. A more direct measure of gene expression is the determination of relative amounts of mRNA by reverse-transcription quantitative PCR (RT-PCR) [86]. Broadening this approach, techniques such as microarrays or RNA-seq represent powerful methods that may allow genome-wide functional transcriptomic analysis [87–89].

### Protein function

When considering the biological role of a gene it is important to understand the function of the protein it encodes, especially if the protein is thought to have an enzymatic function or is likely to interact with binding partners in a wider network. This can include protein–protein interactions, protein–DNA/RNA binding, and other substrate or ligand interactions. Commonly used methods to identify protein-protein interactions are co-immunoprecipitation and protein pull-down assays. These conceptually similar approaches use antibodies to specifically recognise and isolate the (tagged) query protein from a cell lysate, bringing any binding partners with it [90]. Putative binding partners are then identified by mass spectrometry or other similar techniques. Alternative approaches include bacterial two-hybrid systems that are designed to report protein-protein interactions in vivo through expression of a reporter gene, most commonly beta-galactosidase or luciferase. This can be a very rapid approach to discovering or confirming individual predicted interactions, but may also be used to screen libraries of potential protein partners [91, 92].

Classically, DNA- or RNA-protein interactions are identified using DNA/RNA footprinting. Here, protein-bound DNA or RNA molecules are protected from cleavage by nuclease enzymes causing gaps in the digestion patterns when compared to DNA/RNA only controls [93]. More recently, genomic footprinting techniques based on NGS have been established, replacing the final gel separation steps with sequencing [94]. Genome-wide profiles of protein-DNA interactions can be further studied through combining chromatin immunoprecipitation with NGS (ChIP-seq) [95, 96]. Here, the query protein is fixed to the interacting DNA through chemical crosslinking in vivo. These complexes are enriched by immunoprecipitation and, after crosslink reversal, the DNA fragments are released and identified by NGS. ChIP-seq is especially useful for proteins with multiple binding sites, e.g. transcription factors. More focused methods for identifying specific protein-DNA/RNA binding, such as electrophoretic mobility shift assays (EMSA), exploit slower migration rates of protein-nucleic acid complexes in gels compared to nucleic acids alone [97]. Whatever the scale, protein interaction studies are essential for further molecular characterisation of genomic approaches, adding depth and context to candidate genes and advancing our understanding of the bacterial cell.

### Protein localisation

Important clues to the specific function of a protein can be derived by examining its subcellular localisation. Separation of the bacterial cell into simple fractions (such as: membrane, cytoplasm, cell wall) can, in combination with western blotting, give a first indication. However, more specific insights into protein localisation can be achieved by fluorescence microscopy. This method usually involves the creation of protein-reporter fusions and allows tracking of the fluorescent product inside the cell [98–101]. In addition, protein tags, which are made fluorescent through the introduction of a small molecule, have become increasingly popular [102, 103]. While we highlight live-cell microscopy here, the expertise and special equipment required for microscopy methods beyond this quickly scale in complexity with the need for higher resolution.

*Deeper molecular characterisation*

The methods described so far should be practicable in most molecular microbiology laboratories. However, for many laboratories, this is where the research endeavour begins. For example, structural biologists acquire a deeper mechanistic understanding of protein function using methods such as X-ray crystallography [104], nuclear magnetic resonance (NMR) spectroscopy [105] and advanced optical methods such as (cryo-) electron microscopy [106], capitalising on the atomic resolution these methods bring.

## Examples for integrated next-generation microbiology

We have described a direction of travel for next-generation microbiology. Other researchers share the same vision and some studies have begun to bridge the gap and integrate large sequencing datasets with molecular microbiology. For example, early bacterial GWAS made simple comparisons between the putative function of the genes containing associated elements and basic bacterial growth assays with defined substrates [5]. In a more sophisticated approach, and consistent with next-generation Koch's postulates, several studies have used mutagenesis and complementation to test the causality of hits and their predicted phenotype [6, 17, 18, 29, 51, 77]. Going still further, some studies have characterised the function of candidate genes and their genetic context [48, 107, 108], sometimes over multiple publications [84, 109]. For example, combining protein localisation microscopy with protein-protein interaction studies defined the role of candidate genes in the regulation of cell wall biosynthesis [52, 53]. Consistent with this, putative antimicrobial resistance determinants have been investigated in multiple bacterial strain backgrounds to provide information that is increasingly relevant to natural systems [44, 46, 57].

## Future directions

Microbiology research seeks to understand the workings of the bacterial cell in natural environments. Advances in DNA sequencing technologies have touched all aspects of microbiology research but this comes at a price. The specialism required to use these sequencing-based research methods risks a disconnect between bioinformatics and fundamental microbiology. This is because sequencing information is typically viewed through a series of analytical lenses to give it meaning. This means sequencing datasets are frequently abstract in nature and often the mathematical methods used to generate them are challenging to understand for non-specialists. With continued revisions to analytical methods and ever-increasing sample sizes, bioinformatic analysis has the potential to outpace fundamental microbiology investigations by orders of magnitude, exacerbating the analytical disconnect.

There are currently two principal solutions to this problem. First, bioinformaticians can collaborate widely, often contributing specific analytical expertise to each investigation. Second, bioinformatics in some areas has become a specialist data-service where researchers pay for analyses. In both cases, non-specialist researchers are divorced from the data, potentially leaning on internal controls or pre-assumed expectations to guide their interpretations. Similarly, bioinformaticians are removed from deeper understanding and characterisation of their initial discoveries. The typical conclusion is to argue for validated standardised analysis pipelines. This is sometimes necessary, such as in

clinical settings, but in a rapidly evolving field it is vital that researchers are given flexibility for future innovation.

The pace of bioinformatics has moved microbiology research towards the study of natural populations. However, fundamental molecular microbiology approaches continue to focus on laboratory-adapted model strains for consistency across research groups. In the future, it will be important to merge these contrasting approaches to deepen the impact of research studies. Already, association studies (e.g. GWAS) or genome-wide profiles (e.g. Tn-seq) are commonly challenged to validate the gene function of at least one hit, while more fundamental laboratory studies are often compelled to contextualise their findings among more representative strains. In each case, this rising demand places strain on specialist laboratories. Here, we argue for an integrated future for next-generation microbiology, embracing new analysis techniques and placing in silico findings in a microbiological context. The spirit of integrated research is captured by large research collaborations or consortia but integration does not have to be collaboration in the strictest sense, it can be embodied by a small number of individuals who understand complementary research methodologies, provided they find a way to meet the burden of proof set out in next-generation Koch's postulates. With careful reflection on current microbiology practice and a new awareness of the value of communicating complex datasets across disciplines, microbiology has never been in a better position to drive deeper understanding of the natural world.

## Supplementary Information

---

**Additional file 1.** Review history.

---

### Acknowledgements

Icons used in Figs. 1 and 4 are taken from iconfinder.com and used under a creative commons licence and we acknowledge the creators Ivan Boyko, Daniel Bruce, Valera Zvonko, Keris Maker, Yogi Apprelliyanto, PICOL, Graphic Mall and Arana Graphics.

### Peer review information

### Review history

The review history is available as Additional file 1.

### Authors' contributions

CMK, AKF and SKS conceived the review and jointly wrote the paper. The authors read and approved the final manuscript.

### Authors' information

Twitter handles: @carokobras (Carolin M. Kobras); @AndrewKFenton (Andrew K. Fenton); @sheppard_lab (Samuel K. Sheppard).

### Funding

CMK and AKF were funded by Medical Research Council (MRC) grant MR/S009280/1. SKS was supported by MRC grants MR/M501608/1 and MR/L015080/1.

## Declarations

### Competing interests

The authors declare that they have no competing interests.

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

## 
