## [**Additional file 1.** Review history. · Genome Biology]

Review History

First round of review

Reviewer 1

Is the topic of the article timely and of interest to a wide range of readers? Yes

Is the article well written and presented in a logical way? Yes

Do the authors cover the relevant literature in an accurate and balanced way? Yes

Do the authors provide a useful synthesis of the topic? Yes

Do the authors provide insightful discussion of the future directions for the field? Yes

Comments to author:

Kobras et al. discuss a new perspective and roadmap to bridge the ever-increasing genome sequencing data and molecular microbiology techniques to elucidate genotype-phenotype associations and causality. Overall, the review is thoughtful, well-written and timely. It will be of great relevance to bioinformaticians, genome biologists, microbiologists and medical experts.

My main comment is that the proposed roadmap seems more of a list of things to do after GWAS or Tn-seq. The authors proposed that, different microbio techniques are integrated with each other and with GWAS or Tn-seq, especially for those phenotypes caused by multiple independent variations. What is clearly missing are specific examples of studies of this can be or has been done.

Minor comments are listed below.

1. Page 4, lines 101-103: Write in complete sentences.
2. Page 4, line 103: Unclear what "this" in this sentence refers to
3. Page 5, line 118: Insert "with" in "...genetic elements associated (with) complex traits..."
4. Page 6, line 150: The phrase "Outgrowth of these strains provides the selection conditions..." is confusing. It reads as if selection is caused by bacterial growth when it should mean that selection pressure is applied to the bacteria.
5. Figure 3, Next-gen Koch's postulate number 1: Should be "significantly associated" in sentence "The genetic variant under investigation should be (significantly) associated with the phenotype."
6. Page 14: Need to spell out NMR

Reviewer 2

Is the topic of the article timely and of interest to a wide range of readers? Yes

Is the article well written and presented in a logical way? Yes

Do the authors cover the relevant literature in an accurate and balanced way? Yes

Do the authors provide a useful synthesis of the topic? Yes

Do the authors provide insightful discussion of the future directions for the field? Yes. The future directions of the field were clearly addressed.

Comments to author:

In this review Kobras et al discuss the impact of high-throughput sequencing technologies upon current and future approaches to characterising genotype-phenotype associations within bacterial populations. This was an interesting review on an important topic. The review was well-written and enjoyable to read.

1. The review focuses on large bacterial datasets - it may be worth amending the title to make this clear.
2. The authors use the phrase 'DNA sequence datasets' several times e.g. L33. However, they also discuss RNA sequencing e.g. L68 and L292. Could this be changed to 'large sequence datasets'?
3. L94. please expand the acronym SNPs out at first use as this may not be familiar to laboratory based scientists.
4. L222. Minor formatting - please change [Figure 3, 70, 71, 72] to (Figure 3)[refs] to be clear when referring to figures or references.
5. L342. Could a more informative section heading be provided instead of 'Discussion'?

Reviewer 3

Is the topic of the article timely and of interest to a wide range of readers? Yes

Is the article well written and presented in a logical way? Yes

Do the authors cover the relevant literature in an accurate and balanced way? Yes

Do the authors provide a useful synthesis of the topic? Yes

Do the authors provide insightful discussion of the future directions for the field? Yes.

Comments to author:

The review article "Next-generation microbiology: from comparative genomics to gene function" presents some major high throughput sequencing intensive methods and associated bioinformatics for helping to determine genotype to phenotype and possibly more specific gene function. I found the review compelling and pointed out areas for further reading on my part. One could quibble about some tools which were highlighted (GWAS) or not highlighted (pan-genome) but I think it is fair for the authors to streamline their presentation as they see fit. The authors primarily point out the synergy between GWAS (pan-gnome) and TnSeq approaches in a thorough and compelling way. I have recently been involved in comparing some experimental essential gene work (TnSeq) to pan-genome analysis and I found this review informative. The strength of this paper is pointing out possible synergies and highlighting tools and methods across disciplines. The weakness of this paper is more or less hand waving about how to achieve that. Regardless, the paper makes many good points and provides good reading and references. I hope this paper helps pave the way to better cooperation between bioinformaticians and experimentalists or a new crop of better cross-trained researchers but in my experience this is a slow process at best.

RESPONSE TO REVIEWERS

We would like you to address Reviewer 1's main concern about your proposed roadmap and providing examples of studies. We also ask that you include a "Future directions" section, which could comprise either all or part of the section currently called "Discussion."

Consistent with the editors comments we addressed Reviewer 1's main concern as requested (see below) and we re-worked the 'Discussion' as a 'Future Directions' section to emphasize the forward-facing outlook.

Additionally, we have made some minor changes to the abstract of the manuscript, to make it fit the word limit. Please find the edited version pasted at the foot of this email. If you would like to make changes to the title or abstract, please just let us know.

We are grateful for the minor changes to the abstract. These are incorporated within the revised submission.

Please ensure that the manuscript contains the following sections: {Authors' Contributions, Competing Interests, Funding}. You can just rename the "Conflict of interest" section to "Competing interests", and also just make a separate Funding section.

The manuscript now contains the requested sections consistent with the Genome Biology standard format.

Reviewer #1

Kobras et al. discuss a new perspective and roadmap to bridge the ever-increasing genome sequencing data and molecular microbiology techniques to elucidate genotype-phenotype associations and causality. Overall, the review is thoughtful, well-written and timely. It will be of great relevance to bioinformaticians, genome biologists, microbiologists and medical experts.

We are grateful for this positive comment.

My main comment is that the proposed roadmap seems more of a list of things to do after GWAS or Tn-seq. The authors proposed that, different microbio techniques are integrated with each other and with GWAS or Tn-seq, especially for those phenotypes caused by multiple independent variations. What is clearly missing are specific examples of studies of this can be or has been done.

To address this concern, we have now added a section entitled 'Examples for integrated next-generation microbiology' (Line 344-357) to highlight specific example studies that integrate large sequencing data sets with molecular microbiology, either by addressing 'next-generation Koch's postulates' or going even further by investigating the function. We are grateful for the reviewer's suggestion and we think that this addition has improved the manuscript.

Minor comments are listed below.

1. Page 4, lines 101-103: Write in complete sentences.

2. Page 4, line 103: Unclear what "this" in this sentence refers to

These sentences have been revised (Line 99-106); they now read:

'The widespread application of bacterial GWAS has been made possible by adapting the methodological and analytical assumptions of human GWAS in two important ways. First, bacterial GWAS not only targets homologous sequence variation but also aims to identify the numerous accessory genetic elements and genes that may be found in some, but not all, isolate genomes [5, 30]. Second, and most importantly, it accounts for the strong linkage disequilibrium resulting from the clonal mode of bacterial reproduction. Accounting for this population structure is particularly important when considering the genetics underlying phenotype variation as causal variants will be co-inherited with linked loci that may have no adaptive function [12, 13].'

3. Page 5, line 118: Insert "with" in "...genetic elements associated (with) complex traits..."
This has been changed as suggested. (Line 119).

4. Page 6, line 150: The phrase "Outgrowth of these strains provides the selection conditions..." is confusing. It reads as if selection is caused by bacterial growth when it should mean that selection pressure is applied to the bacteria.

This has been revised to improve clarity and accuracy (Line 151) and now reads: 'Selection pressure is applied to these library strains by growing them in defined in vitro or in vivo conditions (Figure 1).'

5. Figure 3, Next-gen Koch's postulate number 1: Should be "significantly associated" in sentence "The genetic variant under investigation should be (significantly) associated with the phenotype."

This has been corrected as suggested (Figure 3).

6. Page 14: Need to spell out NMR

Nuclear magnetic resonance (NMR) has been spelled out (Line 340).

Reviewer #2

In this review Kobras et al discuss the impact of high-throughput sequencing technologies upon current and future approaches to characterising genotype-phenotype associations within bacterial populations. This was an interesting review on an important topic. The review was well-written and enjoyable to read.

We are pleased that, like us, the reviewer sees this as an important topic and welcome their positive comments.

1. The review focuses on large bacterial datasets - it may be worth amending the title to make this clear.
We appreciate the reviewer's suggestion but have preferred to keep the original title. We recognise that the papers we cite are examples, and do not necessarily represent all contributions in the respective area. However, the study is deliberately designed to be a broad, forward thinking review that will be of interest to people working in many areas of microbiology and genomics. With this in mind, we think that the current title describes the content quite well to the prospective reader. Particularly because 'Next Generation Microbiology' deliberately echoes the common genomics term 'Next Generation Sequencing'.

2. The authors use the phrase 'DNA sequence datasets' several times e.g. L33. However, they also discuss RNA sequencing e.g. L68 and L292. Could this be changed to 'large sequence datasets'?

We thank the reviewer for pointing this out. We have now changed this phrase to 'large sequencing datasets' in the manuscript where we refer to general sequencing datasets that could include both DNA and RNA (e.g. L33 and L40).

3. L94. please expand the acronym SNPs out at first use as this may not be familiar to laboratory based scientists.

We now include a definition of this abbreviation immediately before its first use in Line 94.

4. L222. Minor formatting - please change [Figure 3, 70, 71, 72] to (Figure 3)[refs] to be clear when referring to figures or references.

This has now been changed as suggested to improve clarity (Line 223).

5. L342. Could a more informative section heading be provided instead of 'Discussion'?

This has been renamed 'Future directions' in line with the editor's suggestion.

Reviewer #3

The review article "Next-generation microbiology: from comparative genomics to gene function" presents some major high throughput sequencing intensive methods and associated bioinformatics for helping to determine genotype to phenotype and possibly more specific gene function. I found the review compelling and pointed out areas for further reading on my part. One could quibble about some tools which were highlighted (GWAS) or not highlighted (pan-genome) but I think it is fair for the authors to streamline their presentation as they see fit. The authors primarily point out the synergy between GWAS (pan-gnome) and TnSeq approaches in a thorough and compelling way. I have recently been involved in comparing some experimental essential gene work (TnSeq) to pan-genome analysis and I found this review informative. The strength of this paper is pointing out possible synergies and highlighting tools and methods across disciplines. The weakness of this paper is more or less hand waving about how to achieve that. Regardless, the paper makes many good points and provides good reading and references. I hope this paper helps pave the way to better cooperation between bioinformaticians and experimentalists or a new crop of better cross-trained researchers but in my experience this is a slow process at best.

We are delighted with the reviewer's mostly positive comments about the compelling nature of the study. We have addressed the weakness in the revised manuscript to reduce hand waving by incorporating specific examples into a section entitled: 'Examples for integrated next-generation microbiology' (Lines 340-357).